# Voxel-Mirrored Homotopic Connectivity Is Altered in Meibomian Gland Dysfunction Patients That Are Morbidly Obese

**DOI:** 10.3390/brainsci12081078

**Published:** 2022-08-15

**Authors:** Yi-Dan Shi, Hui-Ye Shu, Li-Qi Liu, Shi-Qi Li, Xu-Lin Liao, Yi-Cong Pan, Ting Su, Li-Juan Zhang, Min Kang, Ping Ying, Yi Shao

**Affiliations:** 1Department of Ophthalmology, The First Affiliated Hospital of Nanchang University, Nanchang 330006, China; 2Department of Ophthalmology and Visual Sciences, The Chinese University of Hong Kong, Shatin, New Territories, Hong Kong 999077, China; 3Department of Ophthalmology, Massachusetts Eye and Ear, Harvard Medical School, Boston, MA 02114, USA

**Keywords:** morbid obesity, voxel-mirrored homotopic connectivity, meibomian gland dysfunction, functional connectivity, fMRI

## Abstract

Purpose: To investigate the altered functional connectivity (FC) of the cerebral hemispheres in patients with morbid obesity (MO) with meibomian gland dysfunction (MGD) by voxel-mirrored homotopic connectivity (VMHC). Methods: Patients and matched healthy controls (HCs) were recruited, and all subjects underwent functional resonance magnetic imaging (fMRI), and VMHC results were processed statistically to assess the differences in FC in different brain regions between the two groups. We further used ROC curves to evaluate the diagnostic value of these differences. We also used Pearson’s correlation analysis to explore the relationship between changes in VMHC values in specific brain regions, visual acuity, and Mini-Mental State Examination (MMSE) score. Conclusions: Patients with morbid obesity and MGD had abnormal FC in the cerebral hemispheres in several specific brain areas, which were mainly concentrated in pathways related to vision and perception and may correlate to some extent with the clinical presentations of the patients.

## 1. Introduction

Obesity has become a global health issue. The body mass index (BMI) is widely used in clinical practice to diagnose obesity in patients as well as the degree of obesity. Ideally, individuals should maintain a BMI of 18.5 to 24.9 kg/m^2^ and a BMI of ≥25 kg/m^2^ is defined as obese. When the BMI ≥ 37.5 kg/m^2^, a subtotal gastrectomy to avoid further metabolic disturbances may be necessary [1]. Numerous studies have shown that severe obesity is a clear risk factor for many diseases, including coronary heart disease [2], type 2 diabetes mellitus [3], hypertension [4], and stroke [5]. Obesity is particularly linked to hyperlipidemia [6]. Abnormalities in the lipid metabolism affect all organs and tissues throughout the body, and the relationship between obesity and ophthalmopathy or pathological states of the eye has received a great deal of attention. Obesity is strongly associated with cataracts [7], optic neuropathy [8], age-related macular degeneration [9], high ocular hypertension [10], and diabetic retinopathy [11]. One of the main concerns is the complication of meibomian gland dysfunction (MGD) in severely obese patients, which are characterized by obstruction of the terminal ducts of the lid gland, diffuse lesions, and chronic progression of the disease (Figure 1). MGD is affected by a number of factors, including altered hormone levels [12,13], congenital anatomical abnormalities of the lid gland [14], and chronic blepharitis [15]. Furthermore, a few studies have addressed the relationship between obesity and MGD [16] (Figure 2). A recent large sample multicenter study has provided definitive evidence of the relationship between them [17] and concluded that the association is more pronounced in young people than in the elderly. The mechanisms that establish this link are not yet clear, and the available findings so far have suggested some preliminary hypotheses, notably that: (1) high levels of saturated lipids lead to obstruction of the lid ducts [18], and (2) obesity itself, as a chronic inflammatory disease, also inhibits the anti-inflammatory response, causing elevated levels of pro-inflammatory mediators and free fatty acids in the circulatory system [19]. Further support for this hypothesis is provided by Bu et al.’s study that showed a high-fat diet leads to disruption of corneal endothelial cell pump function and cellular tight junctions [20] and observations in an animal model of excessive lipid accumulation in the lid gland alveoli, obstruction of the ductal orifice, and infiltration of inflammatory cells [21] (Figure 3).

Some fMRI-based studies have shown a degree of cognitive dysfunction in obese patients [22], who may show alterations in functional connectivity (FC) and spontaneous EEG activity in visual, emotional, and cognitive-related brain regions [23,24]. However, it is not clear whether the FC is further altered when excessive obesity is accompanied by MGD. Voxel-mirrored homotopic connectivity (VMHC) is a modern rs-fMRI-based technique that measures the strength of inter-hemispheric FC in corresponding regions of both hemispheres, and it allows assessment of the degree of inter-hemispheric functional synergy [25]. VMHC, amplitude of low frequency fluctuation (ALFF), and functional connectivity density (FCD) are all rs-fMRI-based techniques. However, unlike the other two sequences, VMHC focuses more on the ability to incorporate information exchange and functional interactions between the hemispheres [26]. VMHC has been utilized extensively in the study of a variety of ophthalmic disorders, including strabismus [27], unilateral acute open globe injury [28], corneal ulcer [29], and acute eye pain [30]. It has been demonstrated that the synchrony of neural activity in the visual cortex is closely related to visual perception [31], and the synchrony of it remains well maintained in the early monocularly deprived (MD) model [21].

In line with the above, we aimed to investigate the presence of alterations in VMHC in specific brain regions in obese patients to explore whether these alterations were tied to clinical presentations and potential ocular surface damage.

## 2. Methods

### 2.1. Participants

The inclusion of 12 samples per group in this study was determined based on a sample size of *n* = 15.6 R + 1.6 per group at 80% certainties. Twelve meibomian gland dysfunction patients with morbidly obese (MGD-MO) patients [30] and 12 healthy subjects (HCs) matched for their basic characteristics were identified from the First Affiliated Hospital of Nanchang University and enrolled in this study. Prior to the start of the experiment, basic information is collected on the patient, including visual acuity, blood pressure measured via mercury sphygmomanometer, and cognitive function measured via MMSE score.

All patients incorporated into the MGD-MO group had a BMI ≥ 40 kg/m^2^ and were diagnosed with MGD by a certified clinician. Exclusion criteria for all subjects included: (1) age < 18 years, (2) ocular surface infection from any cause that was in an active phase, (3) ocular disease such as glaucoma, strabismus, and diabetic retinopathy, (4) ocular surgery within six months, (5) medication to control lipid levels or estrogen replacement therapy for any reason, and (6) Alzheimer’s disease, Parkinson’s disease, or otherwise unable to cooperate with the examination.

This study strictly adhered to the Declaration of Helsinki and written approval was obtained from the Ethics Committee of the First Affiliated Hospital of Nanchang University prior to the commencement of the experiment. Professional staff was selected to be in charge of explaining to all subjects the purpose of the study, the content of the experiment, and the potential risks, as well as ensuring that all participants were informed.

### 2.2. MRI Data Collections

The first step was to obtain T1 localization scans, and all subjects underwent MRI to obtain high-resolution T1-weighted images via a three-dimensional spoiled gradient-recalled sequence. The parameters of the scans were: (thickness = 1.0 mm, repetition time = 1900 ms, acquisition matrix 256 × 256, echo time 2.26 ms, field of view: 250 × 250 mm^2^, gap = 0.5 mm, flip angle = 9°). During the eight-minute resting-state scans, the parameters corrected to cover the whole brain were as follows (echo time 30 ms, repetition time approximately 2000 ms, gap = 1.2 mm, thickness = 4.0 mm, acquisition matrix 64 × 64, flip angle = 90°, axial 29, a field of view = 220 × 220 mm^2^). For the duration of the functional MRI scan, all subjects were instructed to stay conscious and avoid deliberate thinking under the guidance of a medical professional, while maintaining their body posture immobile to avoid overthinking.

### 2.3. fMRI Data Preconditioning

Following fMRI scanning of all subjects and T1 images acquisition, data were analyzed with MRIcro software (www.MRIcro.com (accessed on 11 October 2021)), and preliminary data analysis was performed with SPM8 (http://www.fil.ion.ucl.ac.uk/spm (accessed on 11 October 2021)) and DPARSFA (http://rfmri.org/DPARSF (accessed on 11 October 2021)) software to complete the data pre-processing process. Some corrections; i.e., deletions, were made in order to maximize the reflection of the true situation of the experimental data, including: (1) Deleting of the data for the first five time points and only including the data for the remaining 235 time points, due to the possibility of signal instability at the initiation of the examination; (2) Head motor correction: a head immobilizer was used during the examination to minimize displacement of the subject’s head during the examination, requiring a maximum displacement of <1.5 mm in X, Y, and Z directions and a rotation angle < 3° during the scan; (3) time correction; that is, correcting for time differences during the scan to ensure that there was no theoretical difference in the time to acquire voxels at a uniform time point; and (4) spatial standardization: in consideration of the potential for subtle differences in brain volumes between subjects, a standard ultrasound image template was utilized with the study-specific symmetric Montreal Neurological Institute (MNI) as the unifying standard; (5) The images were spatially smoothed using a 6 mm full width half maximum Gaussian kernel; (6) Multiple regression methods were used to eliminate sources of artifacts; (7) Taking into account the effects of low frequency drift and high frequency noise, we improved the accuracy of the results via a temporal filter (0.01–0.08 Hz) [31].

### 2.4. VMHC Data Analyses

The REST software (State Key Laboratory of Cognitive Neuroscience and Learning, Beijing Normal University, Beijing, China) was utilized to bring the data distribution more in line with a normal distribution, specifically through Fisher z-transformation (http://restingfmri.sourceforge.net (accessed on 11 October 2021)), which transforms individual VMHCs into z-values. Individual z-plots were entered into a random effect two-sample *t*-test in a voxelwise manner using the overall VMHC as a covariate to determine the difference in VMHC between HCs and the MGD-MO group. (The statistical threshold was set at the voxel level with *p* < 0.01, alphasim was corrected, and the cluster size was >100 voxels for multiple comparisons.)

### 2.5. Statistical Analyses

All statistical analyses were performed via SPSS 20.0 (SPSS Inc., Chicago, IL, USA). For the section on basic patient information, independent sample *t*-tests were applied to quantitative data, while a chi-squared test was used for qualitative data. Results of VHMC for subjects were first tested for normality, and then independent sample *t*-tests were used for data that conformed to a normal distribution, while a rank-sum test was performed for data that were skewed in distribution. For all statistical analyses, a test level of α = 0.05 was used.

In addition, receiver-operating characteristic (ROC) curve analysis was carried out on brain regions with statistically significant differences in VMHC sequences between the two groups of subjects to assess the clinical value of differences in outcomes across brain regions for disease diagnosis. Second, we examined the patient’s cognitive abilities with the mini-mental state examination (MMSE) scale, and the patient’s visual acuity was also measured. The mean visual acuity of the right and left eyes was taken as the overall mean visual acuity, which was then expressed by logMAR. Finally, we assessed the relationship between subject visual acuity, MMSE score, and VMHC value in brain regions separately using Pearson’s correlation analysis via Origin 2021.

## 3. Results

### 3.1. Basic Information

There was non-statistically significant age difference (*p* = 0.680), gender composition (*p* = 0.356), systolic blood pressure (*p* = 0.417), and diastolic blood pressure (*p* = 0.112) between the two groups, while statistically significant differences were found in left eye visual acuity (*p* < 0.001), right eye visual acuity (*p* < 0.001), MMSE score (*p* < 0.001), and triglycerides (*p* = 0.045; Table 1).

### 3.2. VMHC Analysis Results

We computed VMHC values for both groups of ROIs and identified brain regions with differences in VMHC values between the two groups using statistical analysis. Compared to the HCs, the MGD-MO group exhibited statistically significant decreases in VMHC values in the bilateral inferior temporal gyrus (Temporal_Inf), temporal pole:middle temporal gyrus (Temporal_Pole_Mid), rolandic operculum (Rolandic_Oper), and middle temporal gyrus (Temporal-Mid), while the opposite manifestations occurred bilaterally in the anterior cingulum and paracingulate gyri (Cingulum_Ant) and the precuneus (Table 2, Figure 4).

### 3.3. ROC Analyses

We performed ROC analysis on the brain areas with reduced VMHC for the MGD-MO group compared to HCs to explore the diagnostic value of such alterations in clinical practice. We utilized the area under the curve (AUC) as a measure, as the AUC gives well-balanced sensitivity and specificity of diagnostic indicators. The AUCs we saw in brain regions exhibiting decreased VMHC values in MGD-MO patients were: 0.924 for Temporal-Inf, 0.979 for Temporal-Pole-Mid, 0.965 for Rolandic-Oper, and 0.951 for Temporal-Mid (Figure 5). Therefore, we believe that the VMHC value of Temporal-Pole-Mid has the greatest diagnostic value and can be used as a good indicator to distinguish MGD patients from healthy individuals.

### 3.4. Correlation Analyses

We carried out a correlation analysis for the brain regions with altered VMHC values and the mean visual acuity and MMSE score. MMSE score was positively correlated with the VMHC values in Temporal_Pole_Mid, Temporal_Inf, Rolandic_Oper, Tem-poral_Mid, and negatively correlated with Precuneus. Similarly, LogMAR is negatively correlated with Temporal_Pole_Mid, Temporal_Inf, Rolandic_Oper, Temporal_Mid, and positively correlated with Cingulum_Ant and Precuneus, respectively (Figure 6).

## 4. Discussion

The rising proportion of obese and overweight people has become a global public health issue that is gaining traction. Obesity has reached or exceeded 5% in both males and females in China, and it is still on the rise [32], raising concerns about the health risks of obesity and its complications. We chose to investigate FC changes in specific brain regions in both hemispheres of morbidly obese patients with MGD treated with VMHC.

Our data analysis revealed that VMHC changes in the MO group and HCs were primarily found in the temporal gyrus, with the MO group showing decreased FC in the Temporal-Inf, Temporal-Mid, and Temporal-Pole-Mid areas. Furthermore, the MO group experienced a decrease in VMHC values in Rolandic-Oper, while the exact opposite change was discovered in the cingulum-Ant and precuneus (Table 3).

The temporal gyrus is primarily responsible for processing auditory information and is also thought to be involved in emotion and memory; hence, abnormalities in the anatomical structure or function of the temporal gyrus can often be found in patients with epilepsy [33] or depression [34]. Specifically, Temporal-Inf is involved in a few higher perceptual processes; e.g., visual comprehension [35], and is involved in the composition of the ventral visual pathway and object recognition through vision [36], a suggestion that is strongly supported by atrophy of Temporal-Inf in Alzheimer’s disease [37]. Temporal-Pole-Mid is thought to be involved in the mapping relationship between object naming and auditory comprehension [38], but it has also been found to participate in the later period of eye presence and changes in eye gaze [39]. Temporal-Mid, on the other hand, is predominantly involved in association processes and the construction of novel and useful information [40]. Alterations in the frontal gyrus anatomy, such as spontaneous neuronal activity or FC, have been demonstrated in many diseases. Huang et al. found elevated fALFF values in the inferior temporal gyrus in patients with optic neuritis, presumably as a result of a compensatory DMN network [41]. The MO group also showed a reduction in VMHC values in the middle temporal gyrus and its visual acuity level, and statistical analysis revealed that this association was statistically significant, implying that changes in this area of FC are involved in the MGD development process and may interact with the disease. MGD can cause dry eyes as the disease progresses, and patients frequently exhibit symptoms of ocular discomfort such as photophobia, tearing, and prolonged ocular inflammation [42], but no refractive system pathology is involved, implying that this decrease in visual acuity is transient and reversible, and varies with treatment.

Traditionally, the rolandic operculum was thought to be primarily involved in linguistic processes, such as understanding the speech production [43] and encoding utterance syntax [44]. In patients with schizophrenia, Wu et al. [45] discovered increased FC between the superior temporal gyrus/posterior middle temporal gyrus and the left Rolandic-Oper, implying excessive compensatory activation of the auditory speech-priming function. The significant degeneration of the Rolandic-Oper in amyotrophic lateral sclerosis patients corresponds to the clinical manifestation of their dysarthria [46]. Furthermore, abnormalities in the Rolandic-Oper are thought to be closely linked to the pathogenesis of obsessive-compulsive disorder and the structural basis for impaired language function [47,48]. Zhang and colleagues [49] discovered a negative correlation between blood flow in the Rolandic-Oper and patients’ anxiety levels and visual acuity. Furthermore, it has been proposed that the Rolandic-Oper is heavily involved in the selective processing of conditioned threats in the visual cortex, implying that it is involved to some extent in the functional network of visually relevant pathways [50]. Thus, we discovered that the decrease in VMHC values at the Rolandic-Oper in patients with obesity and MGD may be related to patient anxiety. However, a tractography functional pathway between the Rolandic-Oper and the temporal gyrus has also been discovered [51], indicating that Rolandic-Oper alterations may also be influenced by a decreased FC in the temporal gyrus.

The Cingulum-Ant, a component of the limbic system, is widely thought to play an irreplaceable role in advanced activities, such as cognition [52], memory [53], and attention retention [54], while the dorsal anterior cingulate cortex has also been implicated in visual cognition [55]. Increased ALFF values in the cingulum were observed in patients with unilateral acute open globe injury, and this change correlated with the patient’s age, suggesting a limbic system disorder [56]. Shi et al. [57] hypothesized that the changes in gray matter values and regional homogeneity in patients with advanced monocular blindness were caused by disruptions in the synchronous activity of their neurons. Consequently, rising VMHC values of the Cingulum-Ant suggested that the patient was still in in the early stages of visual impairment and that compensatory FC activity was present in this region.

We eventually discovered that changes in VHMC values also appeared in the precuneus. The precuneus is involved in the composition of the default mode network (DMN) and as well as the composition of the visual network [58]; i.e., in the regulation of visually relevant spatial imaging and visual motion [59]. Many researchers have discovered that the precuneus exhibits grey matter volume atrophy or reduced FC in a variety of diseases, including epilepsy [60], strabismus combined with amblyopia [61], and post-herpetic neuralgia [62]. In contrast, we found that the MO group exhibited elevated precuneus FC, which means that patients may have abnormally hyperactive higher cognitive functions and visuomotor regulation, and we speculate that a compensatory change may occur early in the development of pathological changes in the eyes of the MGD-MO group (Figure 7).

In addition, it is noteworthy that some of the studies found that alterations in functional connectivity were also observed in patients with simple obesity. The study by Zhang et al. [63] observed alterations in the FC of brain regions in obese patients via amplitude of low-frequency fluctuation, but the calculation of the FC in the inferior temporal gyrus in that study yielded results opposite to ours, with the MGD possibly serving as a cause of this discrepancy, but increasing the number of subjects may lead to a different conclusion. Another review summarized research in related fields in recent years and found that the view that obesity is associated with resting-state FC is supported by the results of almost all studies, as evidenced by reduced connectivity in the DMN and emotional management related regions; increased Rs-FC between the hypothalamus and reward areas, the limbic system and the salience network, and reduced with cognitive regions [64]. These findings are broadly consistent with our research results and may further explain why we observed a significantly lower MMSE score in MGD patients than in HCs. We therefore suggest that both MO and MGD may be involved in the alteration of VMHC, and that the exact mechanisms need to be further investigated.

In summary, the alterations in VMHC values in the MGD-MO group were mainly concentrated in visual, language, emotional, and cognitive related brain areas, which may explain the clinical manifestations such as reduced visual and cognitive function that the patients experienced. To our knowledge, this is the first study to explore FC alterations in patients with MGD-MO. In addition, the altered VMHC values in specific brain regions have some value as a diagnostic basis. We also noted the presence of FC alterations in brain regions related to emotional processing and depression, suggesting that it is essential for clinicians to assess depression and anxiety in patients even if they do not present to the clinic with a depressed mood.

Ultimately, it is undeniable that our study still has some weaknesses. To begin with, the participation of too few subjects in the research remains an unavoidable drawback. In addition, as we mentioned earlier, morbid obesity itself may lead to changes in the hemispheric FC of patients, and this should be taken into account when we analyze the changes in VMHC values of patients. Therefore, we believe that further studies with larger sample sizes and more detailed subgroups are warranted.

## 5. Conclusions

MGD-MO patients have been observed to have abnormal hemispheric functional connectivity in brain regions related to eye movements, emotional processing and cognition, and some degree of cognitive decline. This alteration may be caused by MGD, but it cannot be ruled out that obesity itself may be involved. Altered VMHC values may serve as a valuable diagnostic indicator.

## Figures and Tables

**Figure 1 brainsci-12-01078-f001:**
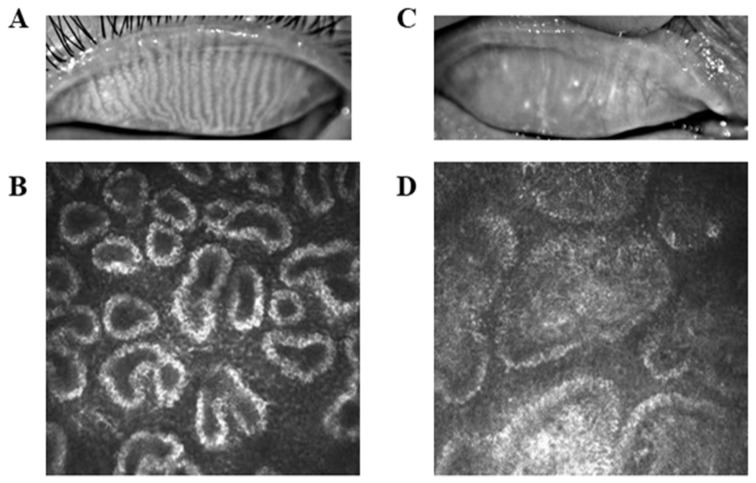
Typical sarcoid manifestations in healthy subjects (**A**) and patients (PAT) (**C**). The ducts of the meibomian gland of the healthy subject are patent as seen by corneal confocal microscopy (**B**), whereas the ducts of the meibomian gland of the PAT appear significantly obstructed (**D**).

**Figure 2 brainsci-12-01078-f002:**
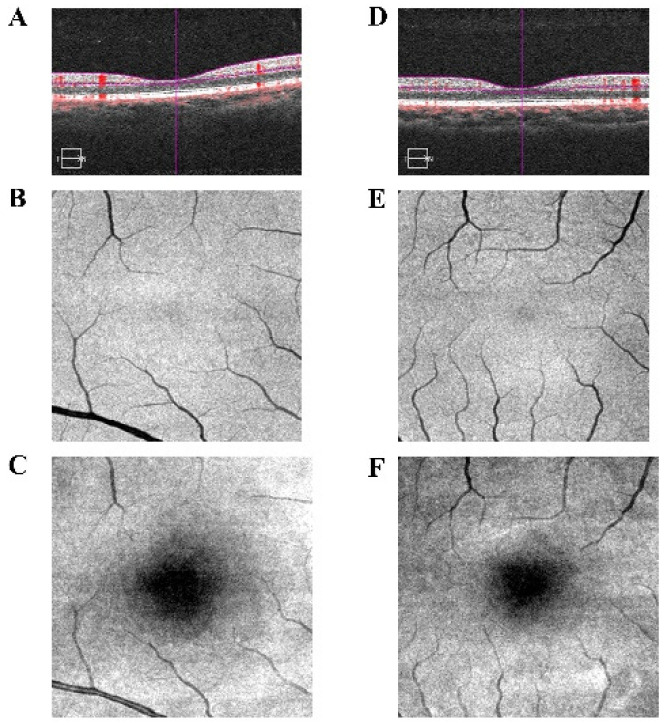
Optical coherence tomography angiography (OCTA) examination of PAT and HCs shows that PAT presented an increase in retinal thickness (**D**) and in fundus vessels (**E**), and a reduction in macular area (**F**) compared to HCs (**A**–**C**).

**Figure 3 brainsci-12-01078-f003:**
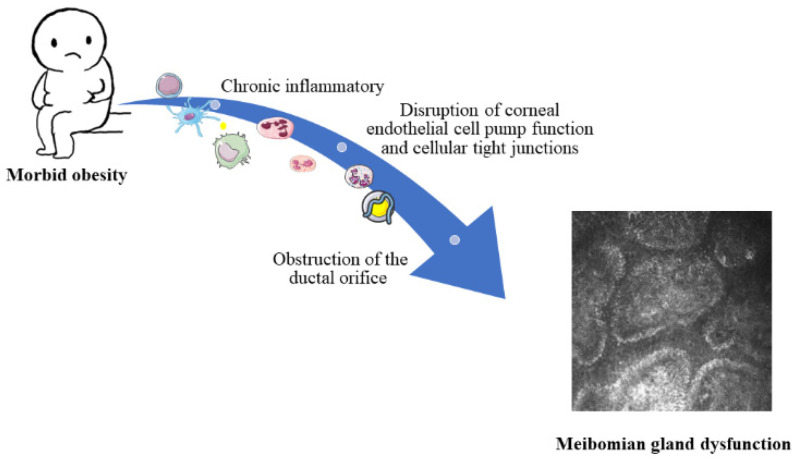
Possible mechanisms of MGD in morbidly obese patients.

**Figure 4 brainsci-12-01078-f004:**
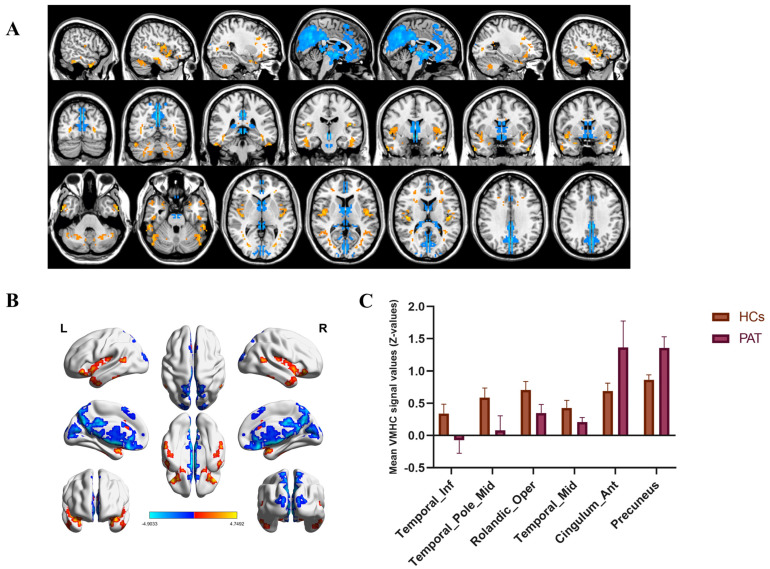
The results of VMHC analysis. Notes: The areas of the brain that show altered VMHC values are labelled on the figure. In (**A**,**B**), red indicates that the PAT group exhibited higher VMHC values than the HCs and the decrease in VMHC values is indicated in blue. The results of the measurement of VMHC values between the two groups, as represented by the bar chart, are presented in (**C**). Abbreviations: HCs, healthy controls; PAT, patients.

**Figure 5 brainsci-12-01078-f005:**
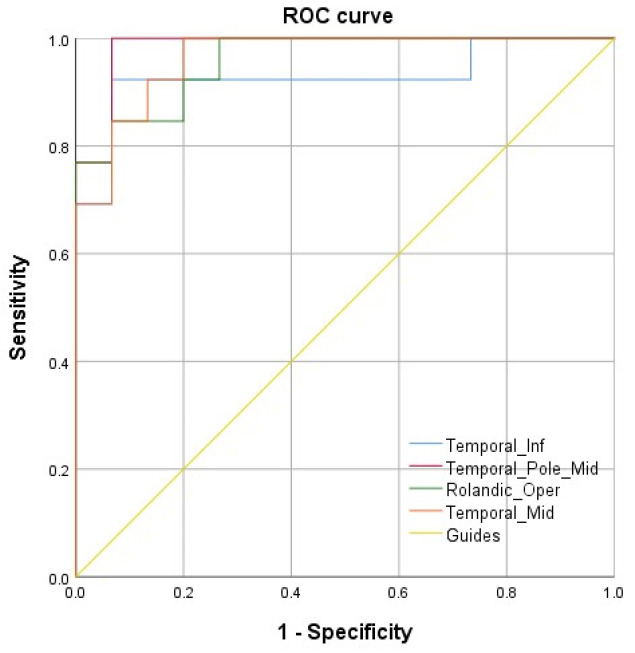
ROC curve. The AUCs were: 0.924 for Temporal-Inf, 0.979 for Temporal-Pole-Mid, 0.965 for Rolandic-Oper, and 0.951 for Temporal-Mid.

**Figure 6 brainsci-12-01078-f006:**
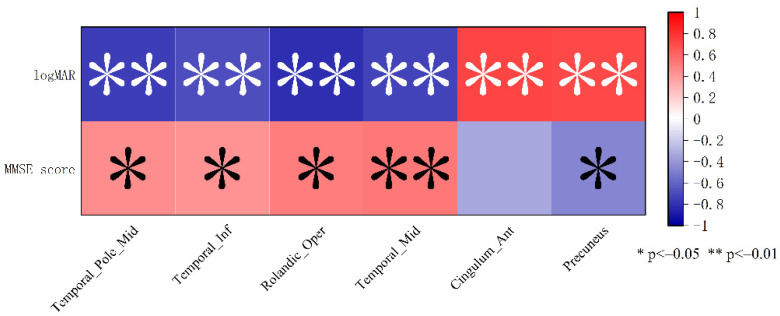
Correlation analysis results. Notes: Red indicates positive correlations; blue indicates negative correlations.

**Figure 7 brainsci-12-01078-f007:**
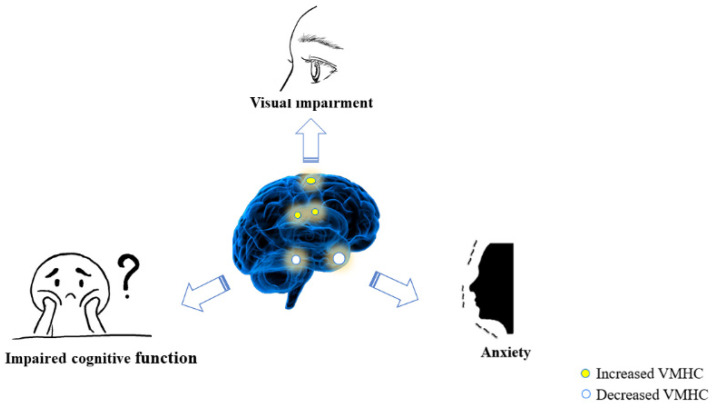
The association between abnormal activity in brain areas and clinical presentation. Notes: Patients exhibit altered VMHC values in specific brain regions that may be associated with a tendency to anxiety, impaired visual acuity, and lower MMSE score.

**Table 1 brainsci-12-01078-t001:** Basic information of all participants.

Condition	PAT	HCs	*t*-Value	*p*-Value
Male/female	4/8	6/6	N/A	0.680
Age (years)	34.25 ± 7.38	31.67 ± 6.24	0.925	0.365
VA-L (log MAR)	0.80 ± 0.17	0.23 ± 3.54	10.972	<0.001
VA-R (log MAR)	0.83 ± 0.23	0.21 ± 0.08	8.867	<0.001
Blood pressure				
SP (mmHg)	126.67 ± 11.85	130.50 ± 10.83	−0.827	0.417
DP(mmHg)	82.41 ± 7.67	76.17 ± 10.57	1.658	0.112
MMSE score	21.42 ± 4.56	27.83 ± 2.52	−4.266	<0.001
TG	2.41 ± 1.20	1.62 ± 0.29	2.231	0.045

Notes: Independent *t*-tests comparing two groups (*p* < 0.05 represented statistically significant differences). Abbreviations: PAT, patient; HCs, healthy controls; VA, visual acuity; N/A, not applicable; SP, Systolic blood pressure; DP, Diastolic blood pressure; MMSE, Mini-mental status examination scale; TG, triglyceride.

**Table 2 brainsci-12-01078-t002:** Brain regions presented alteration VMHC value with statistical significance.

Brain Areas	MNI Coordinates			BA	Peak Voxels	*t*-Value
	X	Y	Z			
HCs > PAT						
Temporal_Inf	−57	−36	−27	37	301	8.85
Temporal_Pole_Mid	57	3	−36	21	153	6.59
Rolandic_Oper	45	−18	15	13	477	6.55
Temporal_Mid	−27	−78	3	22	125	5.29
HCs < PAT						
Cingulum_Ant	3	0	9	32	554	−11.23
Precuneus	−3	−63	33	7	735	−9.58

Notes: The *p* < 0.05 voxel-level statistical threshold for multiple comparisons as using Gaussian random field (GRF) theory (cluster > 100 voxels, FDR corrected; z > 2.3, *p* < 0.01). Abbreviation: VMHC, voxel-mirrored homotopic connectivity; HCs: healthy controls; PAT, patient; MNI, Montreal Neurological Institute; BA, Brodmann area.

**Table 3 brainsci-12-01078-t003:** Brain regions with changed VMHC values and its potential impact.

Brain Region	Brain Function	Prospective Result
HCs > PAT		
Temporal_Inf	Superior perceptual processing; Visual comprehensions; Visual object recognition.	Visual pathway damaged.
Temporal_Pole_Mid	Relationship between object naming and auditory comprehension; Later period of eye presence.	Abnormal eye movement.
Rolandic_Oper	Linguistic process; Utterance syntax encoding.	Possible impairment of speech function; Being influenced by temporal gyrus.
Temporal_Mid	Association processes; Construction of novel and useful information; Selective processing of conditioned threat in the visual cortex.	Impaired cognitive function.
HCs < PAT		
Cingulum_Ant	Advanced activities (e.g., cognition, memory, and attention retention); Visual cognition.	Visual impairment.
Precuneus	Part of DMN; Visual network composition; Regulation of visually relevant spatial imaging and visual motion	Visual impairment.

Abbreviation: HC, healthy controls, PAT, patient, DMN, default mode network.

## Data Availability

The data presented in this study are available on request from the corresponding author. The data are not publicly available due to privacy.

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
