# Peer review of "Voxel-Mirrored Homotopic Connectivity Is Altered in Meibomian Gland Dysfunction Patients That Are Morbidly Obese"

_brainsci, 2022, doi:10.3390/brainsci12081078_

Round 1

Reviewer 1 Report

This paper revealed the altered VMHC in MGD patients. However, before accepting it, it needs to be revised accordingly. Here are my concerns and comments

  1. Abstract: meaningful is a confused word. I guess you mean statistically significant?
  2. Abstract-results is pretty redundant.
  3. Fig1. the definition of PAT is missing.
  4. Fig1 and Fig2: results from study samples or cited? If cited from other studies, please follow journal's requirement.
  5. I appreciated the background information about MGD in introduction. However, the transition from MGD to VHMC technique is not stated clearly. Why do you think VHMC could work? Any supports about the bilateral dysfunction in MGD patients? I suggest to revise this section carefully.
  6. fMRI scan: since conscious would alter FC, please state clearly how to avoid excessive thinking during the scanning. Fixation, watching movie, blank screen or others.
  7. Image registration: which MNI template did you use to perform the study? Symmetric template or asymmetric template?
  8. For VMHC, how did you handle the middle line of brain? These areas have really high VMHC values, which would influence the statistical analysis.
  9. Roc analysis: based on the statement of ROC analysis in method section, this result is not reliable. Using entire dataset to evaluate ROC would cause information leaking. You should, at least, use leave-one-out cross validation.
  10. Results 3.4: why picking middle temporal gyrus? Is this the only region has significant association? And for P values, are they raw values or multiple comparison corrected values?

Author Response

Dear Editors,

Thank you very much for your letter inviting us to submit a revised version of the above-mentioned manuscript. We have revised the paper according to the Reviewer’s suggestions.Specific responses to the reviewers have been submitted in the attachment.

Reviewer 2 Report

Using functional MRI (fMRI) and voxel-mirrored homotopic connectivity (VMHC) technique, Shi et al., investigated the changes in brain functional connectivity in patients with morbid obesity (MO) with meibomian gland dysfunction (MGD). Authors claim that several brain regions, such as, bilateral inferior temporal gyrus, temporal pole:middle temporal gyrus, rolandic operculum, and middle temporal gyrus show significantly decreased VMHC values in MO patients with MGD as compared with the healthy subjects. However, increased VMHC was observed in the bilateral anterior cingulum, paracingulate gyri, and precuneus. Finally, authors conclude that the FC amendments in MO with MGD patients are linked with vision and perception.

Overall, the aim of this study is interesting and has importance in clinical practice. However, there are number of major issues that limit the potentiality of the findings and the authors may consider addressing these concerns prior to a prospective publication.

  1. Relatively small sample size in such clinical study raise major concern on how much confidence one should put on the findings regardless of the advanced imaging and computational pipeline being used. Could authors provide a rationale on this?
  2. The final message of the study was the FC changes in MO-MGD group are linked with the impairment of vision and perception. After going through the discussion section couple times, I had hard time to understand how likely it is! Authors clearly mentioned that ‘Taken together, MO group changes in VMHC in the temporal gyrus suggest that the patient's visual pathways and eye movements may be affected to some degree.’ If the fMRI findings have such trivial affects on vision, I strongly suggest authors to perform some additional measurements, such as, spatial FC mapping of the VMHC along with some diffusion parameters (FA, ADC for example and if possible tractography too) to underpin the structural basis of the observed functional changes. This way authors would be able to convey their message more precisely.
  3. Also, I do not see any strong discussion on their VMHC results with vision and perception and the added clinical values of the study. Authors may consider rephrase the tone of the discussion and highlight the importance of the findings and how these are going to bridge any possible gaps.
  4. It is not clear if the participants were males or females or both?

Author Response

Dear Editors,

Thank you very much for your letter inviting us to submit a revised version of the above manuscript. We revised the paper based on the reviewers' suggestions. Specific responses to the reviewers have been submitted in the attachment.

Round 2

Reviewer 1 Report

I appreciated the revision that was made by the authors. However, some concerns still remain:

1. I don't think keeping posture fixed during MRI scan would help reduce overthinking. 

2. The FC values would be affected a lot by nearby voxels, especially a 6mm spatial smoothing step was applied in this study. I would suggest excluding the middle-line voxels from the analysis since they are too close to each other. 

3. For diagnostic purpose, the AUC should be cross-validated. 

4. If more regions were observed in Result 3.4, you should report all of them. And, please report the results after multiple comparison corrections.

Author Response

Thank you very much for your comment, we have put the document with the responses into the attachment.

Reviewer 2 Report

I would like to thank the authors for addressing my concerns and the manuscript is ready for publication.

Author Response

(The authors gave the same response as above.)
